# Comparative efficacy and safety of caffeine citrate and aminophylline in treating apnea of prematurity: A systematic review and meta-analysis

Yiqun Miao[ID]<sup></sup>, Yun Zhou<sup></sup>, Shuliang Zhao, Wenwen Liu, Aihua Wang[ID]*, Yuanyuan Zhang, Yanan Li, Huimin Jiang

School of Nursing, Weifang Medical University, Weifang, Shandong Province, China

☯ These authors contributed equally to this work.

* wangaihua64@163.com

**Data Availability Statement:** All relevant data are within the paper and its Supporting Information files.

## Abstract

### Background

Methylxanthine, including caffeine citrate and aminophylline, is the most common pharmacologic treatment for apnea of prematurity. However, due to the lack of high-quality evidence, there are no clear recommendations or guidelines on how to choose between caffeine and aminophylline.

### Objective

This meta-analysis aimed to assess the comparative efficacy and safety of caffeine and aminophylline for apnea of prematurity, and provide reliable evidence for clinical medication in the treatment for apnea of prematurity.

### Methods

PubMed, Scopus, Embase, EBSCO, Web of Science, and Cochrane databases were systematically searched from May 1975 to June 2022.

### Results

Ten studies including a total of 923 preterm infants were evaluated. Our results showed that there was no significant difference in the effective rate of 1-3days between caffeine and aminophylline (OR 1.05, 95%CI: 0.40–2.74, $P$ = 0.914). However, for side effects such as tachycardia (OR 0.22, 95%CI: 0.13–0.37, $P$<0.001) and feeding intolerance (OR 0.40, 95%CI: 0.23–0.70, $P$ = 0.001), the incidence rate was lower in the caffeine group compared with the aminophylline group. No significant difference was found in hyperglycemia (OR 0.45, 95% CI: 0.19–1.05, $P$ = 0.064).

**Funding:** The author(s) received no specific funding for this work.

**Competing interests:** The authors have declared that no competing interests exist.

**Abbreviations:** AOP, apnea of prematurity.

## Conclusion

This meta-analysis reveals that caffeine citrate and aminophylline have similar therapeutic effectiveness on respiratory function, but caffeine has fewer side effects and should be considered first for treatment.

## Introduction

Apnea of prematurity (AOP) is a common phenomenon in premature infants, which is defined as the temporary cessation of breathing and may be accompanied by bradycardia and desaturations [1]. This instability of the cardiovascular and respiratory system may be the result of infants' immature respiratory drive and imbalance of sympathetic and parasympathetic outputs [2]. The incidence of AOP is negatively related to gestational age, with 85% of infants less than 34 weeks of gestational age developing AOP and almost all infants less than 30 weeks of gestational age or less than 1000g of birth weight developing AOP [3]. If AOP persists or is poorly controlled, it can lead to the development of bronchopulmonary dysplasia and retinopathy of preterm infants, and even increase infant mortality [4]. Therefore, considering the potential short-term and long-term effects of apnea, early and effective clinical intervention is vital.

Although many options exist for treating apnea in preterm infants, there is no clear "first-line" approach or gold standard of care. At present, methylxanthine therapy is the mainstay of treatment for apnea, including caffeine and aminophylline/theophylline [5]. Despite its widespread use, there is still considerable uncertainty in the efficacy and safety of treating apnea of prematurity [6]. Several studies have proved that caffeine has better therapeutic efficacy with fewer adverse effects, is more reliably absorbed when administered enterally, and is gradually replacing aminophylline as the primary drug for AOP [7, 8]. However, a previous study assessed that the positive effects of caffeine on premature infants' respiratory may be offset by its negative effects on the central nervous system [9]. Recent studies have also shown that caffeine citrate increased the metabolism and oxygen consumption, which could lead to irritability in children [10, 11]. And caffeine citrate is expensive, there are often shortages of the drug in some parts of the world, aminophylline is still the most widely used medicine especially in developing countries [12]. Therefore, there are still some controversies surrounding the choice between caffeine citrate and aminophylline for apnea of prematurity.

Due to the limited hepatic, renal, and respiratory functions of the preterm infants, there is great concern about choosing the one with quick effects, small side effects and low toxicity in the course of treatment. Therefore, this meta-analysis focuses on reliable data regarding the efficacy and safety of caffeine and aminophylline for apnea of prematurity, aiming to provide effective evidence-based medication guidelines for clinicians.

## Materials and methods

### Systematic search and strategy

Our research team performed the meta-analysis based on the guidelines of the Preferred Reporting Items for Systematic Reviews and Meta-analysis (PRISMA) [13]. PRISMA checklist was shown in S1 Table. We searched Embase, PubMed, Web of Science, EBSCO, Cochrane, and Scopus databases from May 1975 to June 2022 for all relevant articles. The search was conducted based on the 5 elements of the PICOS model, 2 of which (population, intervention/

exposure) were addressed in the search strategy (S2 Table). The search strategy combined two areas as MeSH terms, keywords, and text words using Boolean operators: ("Caffeine Citrate" OR "Caffeine" OR "Citrates") AND ("Theophylline" OR "Aminophylline" OR "Diaphylline" OR "Aminophyllinum") AND ("Apnea" OR "Apnoea") AND ("Preterm infants" OR "Infant, Premature" OR " Premature Infant " OR " Infants, Preterm" OR " Neonatal Prematurity").

## Inclusion and exclusion criteria

In this meta-analysis, the research needed to meet the following inclusion criteria:

1. Infants born <37 weeks of gestation and diagnosed with apnea (Respiratory arrest for $\geq$20 s, may be accompanied by bradycardia (heart rate <100 bpm) or oxygen desaturation ($SpO_2$ <80%), and that occurred more than 12 h after birth.);

2. Randomized controlled studies and observational studies;

3. A group exposed to caffeine citrate therapy (exposure group), a group exposed to aminophylline therapy (control group), and there was a comparison among the two groups;

4. At least one of the parameters was included in the outcome measures: the effective rate of 1-3days, sinus tachycardia, feeding intolerance, hyperglycemia, total adverse effects. The effective of 1-3days was defined as 50% or greater reduction in apnea episodes within 72h and <3 times/day after administrating the drugs [14]; Sinus tachycardia is defined as the heart rate persistence >160 bpm or the heart rate > 20% of the baseline heart rate after medication in infants. The diagnosis of feeding intolerance and hyperglycemia relies on the documentation of medical or surgical data.

In this meta-analysis, any of the following literature that met the exclusion criteria were excluded:

1. Reviews, case reports, meeting abstracts, animal trials, or trial registration reports;

2. Full text was unavailable;

3. Unclear or inappropriate definition of exposure/results;

4. Limited information was available for extracting or duplicating data.

## Data extraction

Two researchers reviewed each study independently according to the inclusion and exclusion criteria, any ambiguities were reviewed by a third researcher. First, we eliminated duplicate files, filtered the studies by titles and abstracts, and then read the full-text details to filter irrelevant studies. The two researchers recorded with excel spreadsheets and independently extracted data from articles that met the requirements. The specific format is as follows: (1) General characteristics of study: study, year, study type. (2) Intervention: gestation age, the number of participants, dosage. (3) Outcomes.

## Quality assessment

The research team used the Newcastle-Ottawa Scale (NOS) to assess the risk of bias in cohort studies, with three factors of patient selection, group comparability, and outcome assessment. Higher scores represent higher quality, and more than 7 out of 9 represent high-quality articles. We also used Jadad scale (ranging from 0 to 5) to evaluate the risk of bias in randomized

controlled trials, including randomized evaluation, blind method of outcome evaluation and description of withdrawal and dropouts. A study with a Jadad score of more than 3 was considered to be of high quality.

## Statistical analysis

In our study, STATA software version 12.0 (Stata Corp, College Station, TX) was used for analysis. For continuous results, MD and 95% confidence intervals (CI) were used for calculations. For dichotomous results, OR and 95% CI were used for calculations. Heterogeneity was evaluated by $I^2$ statistics. When $I^2 < 50\%$, no heterogeneity was considered to exist. When $I^2 \geq 50\%$, a great heterogeneity was considered. Funnel plot, Begg's and Egger's tests were performed to evaluate publication bias. To evaluate the stability of the results, sensitivity analysis of ignoring one study at a time was carried out. $P < 0.05$ was considered to be statistically significant.

# Results

## Search results

A total of 408 studies were initially screened by the research team, with 373 remaining after duplicate publications were eliminated by computer. Then, 326 studies were excluded by screening titles and abstracts. Finally, two randomized controlled trials and eight observational studies were considered eligible for inclusion [15–24]. The flowchart depicts the selection procedure for the ten studies (Fig 1). Table 1 demonstrates the characteristics of the included studies.

S3 Table shows the details of the NOS quality assessment. From the Newcastle-Ottawa scale, the quality of the included cohort studies ranged from 7 to 9, indicating that the quality of the studies was fair. The quality of the 2 randomized controlled trials we included was at an intermediate level.

## Meta analysis results

**The effective rate of 1-3days.** The effective rate of 1-3days was included as an outcome of 5 studies which involved 365 infants. No significant difference was observed between the caffeine group and the aminophylline group regarding the effective rate of 1-3days (OR 1.05, 95% CI: 0.40–2.74, $P = 0.914$). There was significant heterogeneity between these studies, so we performed the randomed-effect model in our study ($I^2 = 51.6\%$, $P = 0.083$, Fig 2).

**Tachycardia.** Tachycardia was included as an outcome between caffeine citrate and aminophylline in 6 studies which involved 557 preterm infants. In the meta-analysis, we observed that the caffeine group had lower incidence of tachycardia compared with the aminophylline group (OR 0.22, 95%CI: 0.13–0.37, $P < 0.001$). There was no significant heterogeneity among these studies ($I^2 = 0.0\%$, $P = 0.588$, Fig 3). Thus we conducted the meta-analysis using a fixed-effect model.

**Feeding intolerance.** A total of 4 studies (363 infants) recorded the number of infants with feeding intolerance in the caffeine group and the aminophylline group. Compared with aminophylline, preterm infants treated with caffeine citrate had a lower probability of feeding intolerance (OR 0.40, 95%CI: 0.23–0.70, $P = 0.001$). Heterogeneity among studies was low ($I^2 = 38.6\%$, $P = 0.181$, Fig 4).

**Hyperglycemia.** Hyperglycemia was included as an outcome between caffeine citrate and aminophylline in 3 studies which involved 343 preterm infants. There was no significant heterogeneity between these studies ($I^2 = 0.0\%$, $P = 0.617$, Fig 5). However, there was no

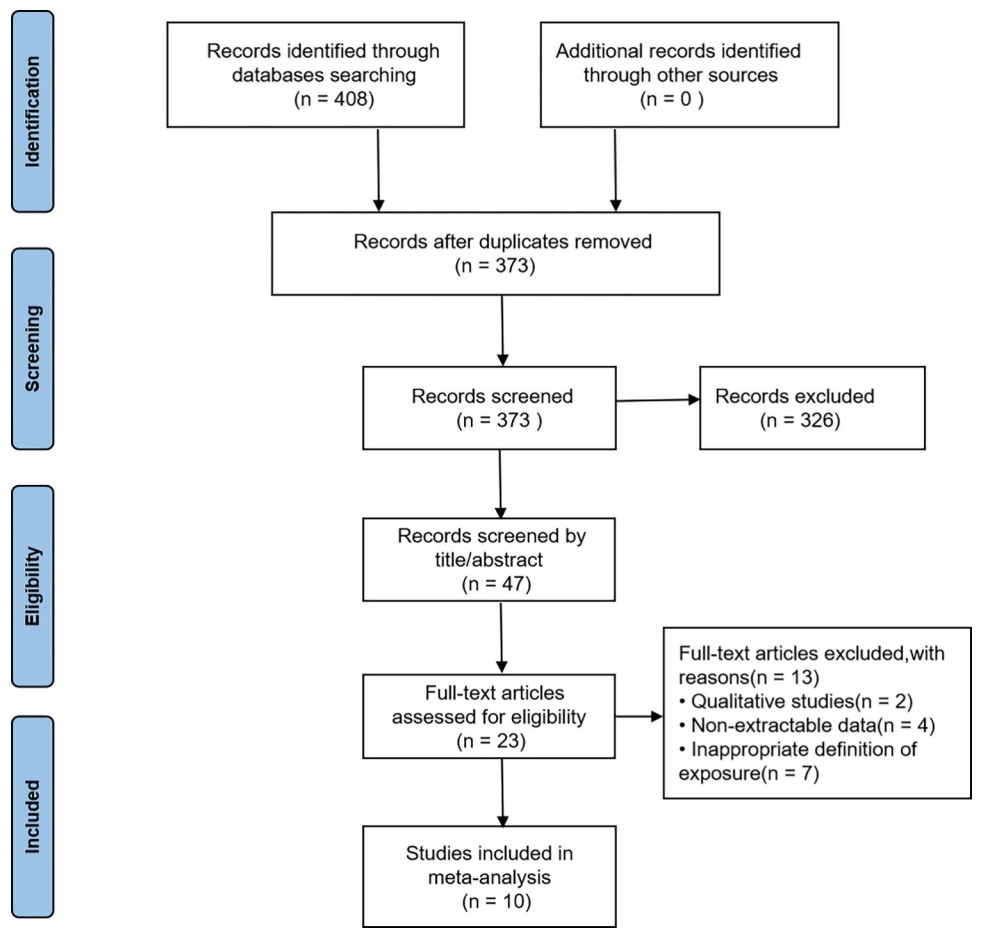

**Fig 1. Flow chart of study selection process.**

significant difference between the caffeine citrate and aminophylline groups in terms of hyperglycemia (OR 0.45, 95%CI: 0.19–1.05, $P$ = 0.064).

**Total adverse effects.** Total adverse effects was included as an outcome between caffeine citrate and aminophylline in 4 studies which involved 128 preterm infants. The analysis found that compared with aminophylline, infants in the caffeine group had a lower chance of developing total adverse effects (OR 0.41, 95%CI: 0.17–0.97, $P$ = 0.042). And the heterogeneity among studies was low ($I^2$ = 22.4%, $P$ = 0.276, Fig 6).

## Sensitivity and publication bias analysis

Sensitivity analysis of each indicator showed that the results were reliable and stable. Publication bias was calculated using the funnel plot, and there was no obvious asymmetry. Moreover, no obvious publication bias was detected by Begg's and Egger's tests ($P > 0.05$).

## Discussion

Although methylxanthine has been widely administered in many countries for the treatment of AOP, the choice between caffeine and aminophylline is still controversial. It is well known that caffeine has a wider effectiveness as well as less adverse effects than aminophylline, and does not require routine blood concentration measurements [25]. But the present findings

**Table 1. Characteristics of included studies.**

| Study | Year | Study type | Gestation age(weeks) | | Participants | | Dosage | | Outcomes | NOS/JS |
|---|---|---|---|---|---|---|---|---|---|---|
| | | | caffeine | aminophylline | caffeine | aminophylline | caffeine | aminophylline | | |
| Lin | 2022 | Cohort study | 30.08 ± 2.73 | 29.96 ± 2.57 | 48 | 96 | iv, the loading dose was 20 mg/kg, followed by a maintenance dosage of 5 mg/kg/dose once per day. | iv, the loading dose was 5 mg/kg, followed by a maintenance dosage of 3 mg/kg/day divided into three doses a day | ② | 7 |
| Zhang | 2020 | Cohort study | 29.78±1.42 | 31.44±1.20 | 77 | 43 | iv, at a first dose of 20 mg/kg followed by a maintenance dose of 10 mg/kg per day | iv, at a first dose of 5 mg/kg followed by a maintenance dose of 2.5 mg/kg twice per day | ① | 7 |
| Nagasato | 2018 | Cohort study | 23.9–34.1 | 23.7–35.9 | 34 | 28 | iv, a maintenance dose of 6.4mg/kg/d iv for 47 days | iv, a maintenance dose of 3.2 mg/kg/d for 42 days | ①③④⑤ | 7 |
| Shivakumar | 2017 | Cohort study | 29.4±2 | 29.3±1.9 | 77 | 79 | iv or oral, the loading dose was 20 mg/kg and the maintenance dose of 5 mg/kg 24 hourly | inj, the loading dose 5 mg/kg and maintenance dose of 1.5 mg/kg 8 hourly | ②④ | 8 |
| Xu | 2014 | Cohort study | 26–36 | 26–36 | 65 | 60 | iv, at a first dose of 20 mg/kg followed by a maintenance dose of 5 mg/kg per day | iv, at a first dose of 5 mg/kg followed by a maintenance dose of 2 mg/kg 12 hourly | ②③④ | 7 |
| Skouroliakou | 2009 | Cohort study | 30.4±1.9 | 31.5±1.6 | 33 | 37 | the loading dose was 20 mg/kg given intravenously over 30 min and the maintenance doses was 5 mg/kg per day | iv, the loading dose was 4.8 mg/kg and the maintenance doses was 2 mg/kg every 12 h | ① | 9 |
| Larsen | 1995 | Cohort study | 25–33 | 26–33 | 82 | 98 | iv loading dose of caffeine citrate 20.2 mg/kg followed by a maintenance dose of 2.5mgIkg iv or by a gastric baby feeding tube twice a day for 10 days. | the loading dose of 6.2mg/kg followed by a maintenance dose of 3.1 mg/kg iv or by a gastric baby feeding tube twice a day for 10 days. | ① | 9 |
| Scanlon | 1992 | Cohort study | 28.7±1.2 | 27.9±1.4 | 16 | 14 | ivgtt, at a first dose of 25 mg/kg followed by a maintenance dose of 6 mg/kg per day | ivgtt, at a first dose of 7.5 mg/kg followed by a maintenance dose of 3 mg/kg 8 hourly | ①②⑤ | 7 |
| Bairam | 1987 | RCT | 30.0±1.5 | 30.3±0.8 | 10 | 10 | ivgtt, at a first dose of 10 mg/kg followed by a maintenance dose of 1.25 mg/kg 12 hourly | ivgtt, at a first dose of 6 mg/kg followed by a maintenance dose of 2 mg/kg 12 hourly | ①③⑤ | 3 |
| Brouard | 1985 | RCT | 30.5±0.7 | 30.5±0.4 | 8 | 8 | ivgtt, at a first dose of 20 mg/kg followed by a maintenance dose of 5 mg/kg 24 hourly | ivgtt, at a first dose of 5.5 mg/kg followed by a maintenance dose of 0.5–2.5 mg/kg 8 hourly | ①⑤ | 3 |

Notes:①the effective rate of 1-3days②tachycardia③feeding intolerance④hyperglycemia⑤total adverse effects.

reported that there is no difference between caffeine and aminophylline in terms of incidence apneic, bradycardic or hypoxemic [26]. What's more, aminophylline is effective in improving the renal function of asphyxiated neonates [27]. Therefore, our meta-analyses aimed to assess the comparative efficacy and safety of caffeine and aminophylline in the treatment of AOP. The results indicated that caffeine was as effective as aminophylline in terms of treating AOP, the occurrence of tachycardia and feeding intolerance in the caffeine group was clearly less than that in the aminophylline group. But the rates of hyperglycemia were similar between the caffeine and aminophylline groups.

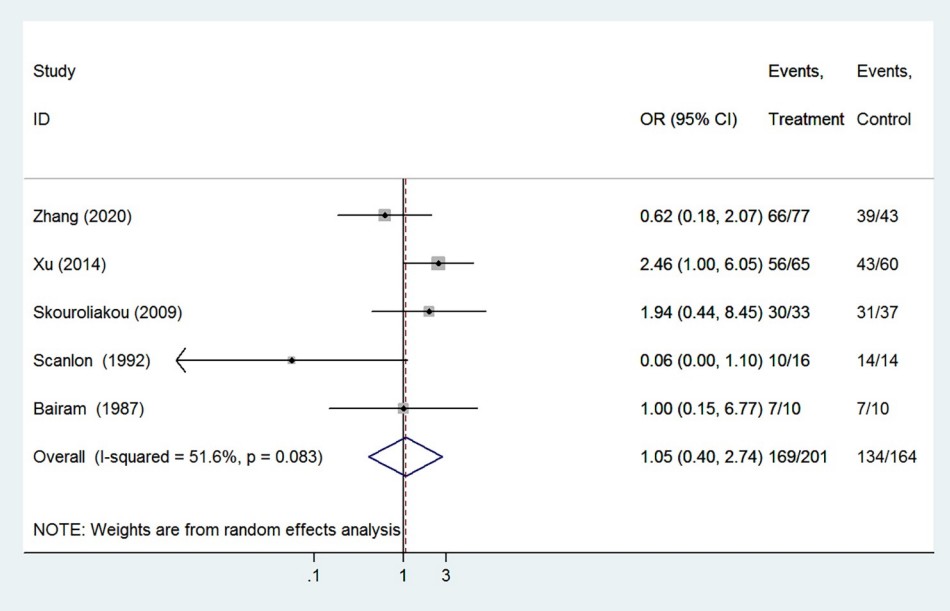

**Fig 2. Forest plot of showing the effects of caffeine vs aminophylline on the effective rate of 1–3 days.**

We believe that caffeine and aminophylline treatments are equivalent effective in reducing apnea of prematurity, which has been confirmed in several independent studies [28, 29]. Steer similarly concluded that in the short term, caffeine was as effective as aminophylline in reducing the incidence of apnea and was better tolerated and easier to administer. These are related to the mechanism of methylxanthine, caffeine and aminophylline are both central nervous

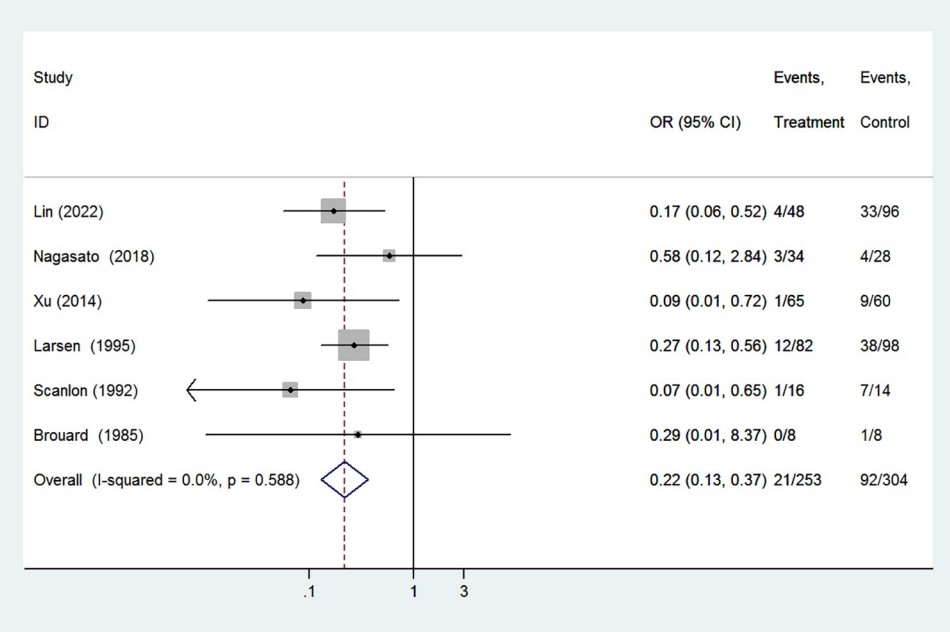

**Fig 3. Forest plot of showing the effects of caffeine vs aminophylline on tachycardia.**

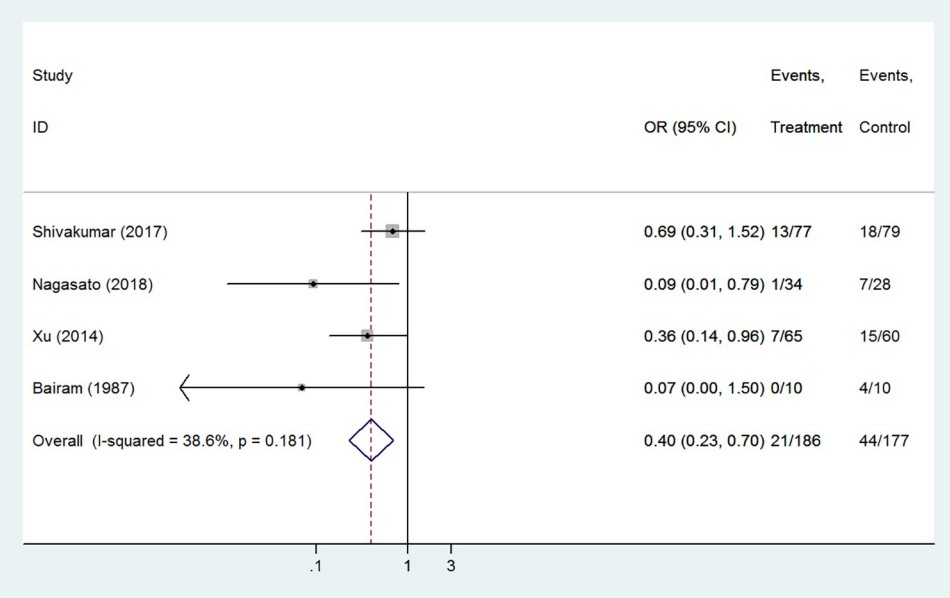

**Fig 4. Forest plot of showing the effects of caffeine vs aminophylline on feeding intolerance.**

system stimulants. They have the same effect of increasing diaphragm contractility and lowering hypercapnia susceptibility threshold [30]. Some studies have further shown that prophylactic use of caffeine for preterm infants at risk of apnea reduced the duration of mechanical ventilation [31]. Borszewska-Kornacka [32] found that preventive use of caffeine within 24 hours after birth can reduce the use of invasive ventilator and the incidence of IVH and PDA compared with therapeutic use of caffeine. However, whether caffeine should be used

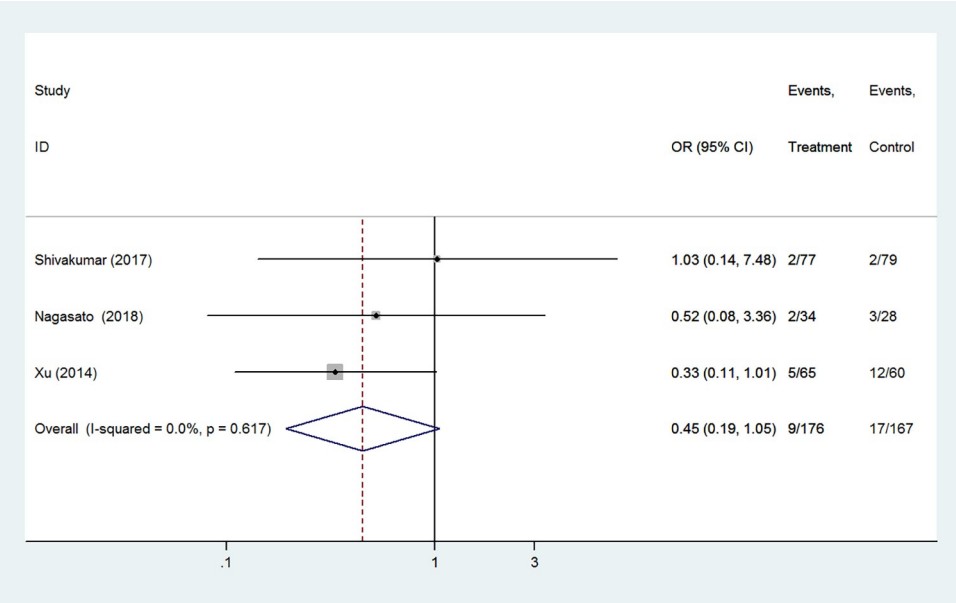

**Fig 5. Forest plot of showing the effects of caffeine vs aminophylline on hyperglycemia.**

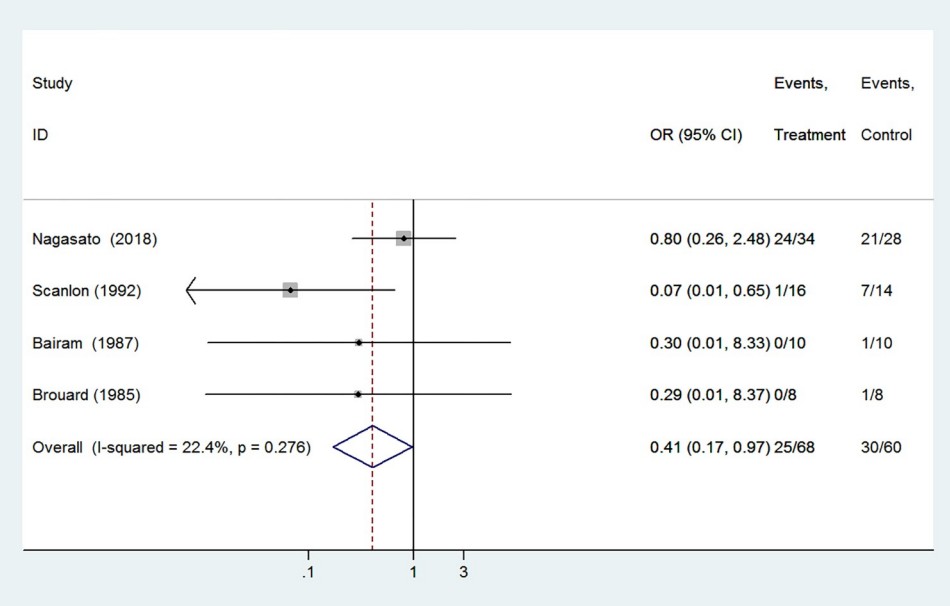

**Fig 6. Forest plot of showing the effects of caffeine vs aminophylline on total adverse effects.**

prophylactically in all preterm infants remains controversial [33]. It is necessary to conduct a definitive study to illustrate the comparative effects of prophylactic and therapeutic caffeine.

Tachycardia is one of the most common acute adverse reactions of methylxanthines. We demonstrated a lower incidence of tachycardia in infants treated with caffeine compared with aminophylline. Our result is consistent with those obtained by Alhersh et al. [34]. One possible interpretation of the result is that caffeine is more active than aminophylline in stimulating central nervous system, but less active in stimulating heart [35]. In addition, aminophylline contains phenethyl alcohol, which could directly stimulate the respiratory center to increase its sensitivity to carbon dioxide, leading to an increase in respiratory rate, thus causing tachycardia in preterm infants. Ultimately, aminophylline exhibits unique toxicokinetics in premature infants with sinus tachycardia due to the great developmental differences between neonates and adults [36]. As a result, caffeine is superior to aminophylline in terms of the risk of tachycardia.

Our study showed that caffeine significantly reduced the incidence of feeding intolerance compared with aminophylline treatment, the result is indeed in line with research of Rostas et al. [4]. Analyzing the possible causes, the serum half-life of aminophylline ranges from 24.7 to 36.5 h, while caffeine's serum half-life is 101 h in preterm infants [18]. Caffeine has a longer elimination half-life, allowing for once-daily dosing. Therefore, administration once a day has less effect on the upper gastrointestinal tract than administration multiple times a day. Another reason could be that caffeine is absorbed more rapidly by gastrointestinal tract than aminophylline, with little or no first-pass metabolism [37]. In addition, the dosage of caffeine therapy can be challenging because of the limited hepatic, renal, and respiratory function of preterm infants. A previous study has shown that high doses of caffeine lead to the decrease of mesenteric blood flow and serious feeding intolerance of caffeine generally appears at concentrations >60 mg/L [30]. But chen [38] found that higher doses of caffeine (10–20 mg/kg daily) were more effective than lower doses (5–10 mg/kg daily) and that higher doses were more closely associated with withdrawal from the ventilator. Definitive recommendations on the optimal

caffeine dose cannot be given due to our research not making a specific distinction between drug doses, future research is necessary to elucidate the effects of different maintenance doses of caffeine in the treatment of AOP.

Furthermore, our study shows the risk of hyperglycemia of caffeine is similar to that of aminophylline. Our results are different from those of Hady et al., who found caffeine has little risk of hyperglycemia, which is associated with its hydrophobicity and distributes rapidly without tissue accumulation [35]. The discrepancy between our study and others may be due to the insufficient number of studies we included on measures of hyperglycemia. More well-designed and larger sample studies should be carried out to examine and expand the conclusions.

Some literature did not describe specific types of adverse reactions, but summarized all the adverse reactions in general, so we further discussed the risk of total adverse effects. The results of our meta-analysis indicated that the total adverse effects rate of caffeine in the treatment of apnea is lower than that of aminophylline. It may be related to the narrow safe range of aminophylline concentration in blood. Ye [39] have also suggested that excessive serum concentration of aminophylline was related to an increased risk of adverse effects. Patients with aminophylline blood concentration >20 μg/ml may have symptoms such as tachycardia and arrhythmia, while patients with aminophylline blood concentration >40 μg/ml may have symptoms such as fever, dehydration, convulsion and even death. However, caffeine has a wide range of safety, with serum levels of 3–84 mg/L considered therapeutic and safe [40]. Furthermore, the plasma concentration fluctuation of caffeine in neonates is smaller than aminophylline, caffeine has greater central nervous system penetration, and does not produce fluctuation of cerebral blood flow [41]. Liu et al. [42] has shown that caffeine may have a direct long-term neuroprotective effect on the developing brain. Mürner-Lavanchy et al. [43] also reported that caffeine therapy for apnea of prematurity improved visuomotor, visuoperceptual, and visuospatial abilities at age 11 years. General intelligence, attention, and behavior were not adversely affected by caffeine. In order to truly prove the safety of caffeine citrate, more long-term follow-up studies are necessary.

Our meta-analysis differs from other analyses in that it not only compares the effectiveness of caffeine and aminophylline, but also evaluates their safety in preterm infants. Despite the overwhelming evidence that caffeine is safer and more effective than aminophylline, and the resulting widespread use in developed countries. However, the use of caffeine therapy remains alarmingly low in sub-Saharan Africa, greater efforts are needed to ensure broader utilization of caffeine therapeutic regimens.

## Limitation

It is worth noting that our meta-analysis still has many limitations. First, limited number of RCTs were retrieved, and we included relevant non-RCT studies to expand the total sample size. The limited number of RCTs resulted in a smaller randomized sample size, which may have biased the results. Moreover, the information on long-term outcomes of methylxanthines in our included studies is still insufficient and large-scale trials should be conducted in the future to evaluate the potential long-term adverse outcomes in clinical practice.

## Conclusion

In summary, evidence has shown that caffeine and aminophylline are equally effective for apnea of prematurity but caffeine has fewer adverse effects on infants. For these reasons, we recommend caffeine as an effective and feasible first-line treatment for apnea in preterm infants. We believe it will help update the available evidence on the benefits and harms of

methylxanthine treatment for AOP, thus enabling researchers and clinicians to make the proper choice for the treatment of AOP.

## Supporting information

**S1 Table. PRISMA checklist.**
(DOCX)

**S2 Table. Search strategy.**
(DOCX)

**S3 Table. Assessment of methodological quality by NOS.**
(DOCX)

## Author Contributions

**Conceptualization:** Yiqun Miao, Yun Zhou, Shuliang Zhao, Huimin Jiang.

**Data curation:** Yiqun Miao, Yun Zhou, Shuliang Zhao, Wenwen Liu, Yanan Li.

**Formal analysis:** Yiqun Miao, Yun Zhou, Shuliang Zhao.

**Funding acquisition:** Aihua Wang, Yuanyuan Zhang.

**Investigation:** Yiqun Miao, Yun Zhou, Shuliang Zhao, Wenwen Liu, Huimin Jiang.

**Methodology:** Yiqun Miao, Yun Zhou, Shuliang Zhao, Wenwen Liu, Yanan Li, Huimin Jiang.

**Project administration:** Aihua Wang, Yuanyuan Zhang.

**Resources:** Aihua Wang.

**Software:** Yiqun Miao, Yanan Li.

**Validation:** Shuliang Zhao, Wenwen Liu, Aihua Wang, Yuanyuan Zhang.

**Writing – original draft:** Yiqun Miao, Yun Zhou.

**Writing – review & editing:** Yiqun Miao, Aihua Wang, Yuanyuan Zhang.

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
