## [Decision Letter · Decision Letter 0]

9 Aug 2022

PONE-D-22-17592Comparative efficacy and safety of caffeine citrate and aminophylline in treating apnea of prematurity: A meta-analysisPLOS ONE

Dear Dr. Wang,

Thank you for submitting your manuscript to PLOS ONE. After careful consideration, we feel that it has merit but does not fully meet PLOS ONE’s publication criteria as it currently stands. Therefore, we invite you to submit a revised version of the manuscript that addresses the points raised during the review process.

We look forward to receiving your revised manuscript.

Kind regards,

Hasan Tolga Celik, M.D.

Academic Editor

PLOS ONE

Journal Requirements:

2. Please provide the date range over which the literature search was conducted. Specifically, please provide the date of the first and the last literature search.

3. Please amend your title to include "systematic review and meta-analysis".

Reviewers' comments:

Reviewer's Responses to Questions

**Comments to the Author**

1. Is the manuscript technically sound, and do the data support the conclusions?

Reviewer #1: Yes

Reviewer #2: Yes

2. Has the statistical analysis been performed appropriately and rigorously? 

Reviewer #1: Yes

Reviewer #2: Yes

3. Have the authors made all data underlying the findings in their manuscript fully available?

Reviewer #1: Yes

Reviewer #2: Yes

4. Is the manuscript presented in an intelligible fashion and written in standard English?

Reviewer #1: Yes

Reviewer #2: Yes

5. Review Comments to the Author

Reviewer #1: Methylxanthines, including caffeine citrate and aminophylline, is the most common pharmacologic treatment for apnea of prematurity. However, due to the lack of high-quality evidence, there are no clear suggestions or guidelines on how to

choose between caffeine and aminophylline. This meta-analysis aimed to assess the comparative efficacy and safety of

caffeine and aminophylline for AOP, and provide reliable evidence for clinical medication in the treatment for apnea of prematurity. This meta-analysis reveals that caffeine citrate and aminophylline have similar therapeutic effectiveness on respiratory function, but caffeine has fewer side effects and should be considered first for treatment.

So this research is a nice work.

Yours sincerely,

Reviewer #2: Dear Authors,

The study is well designed, writing and grammar are clear.

There were limited RCTs in the study and it’s difficult to recommend the optimal treatment.

Early caffeine initiation is recommended in the literature.

In a study, early caffeine initiation (initial dose on the 1st day of life) was associated with a decreased need for invasive ventilatory support and lower incidence of IVH and PDA (PLoS ONE 12(12): e0189152. https://doi.org/10.1371/journal.pone.0189152X) that they recommend the initiation of caffeine treatment in the first 24 hours of life. The initiation time of caffeine treatment and its relationship with adverse effects can be mentioned in the discussion section.

The dose of caffeine treatment can be challenging because of the limited hepatic, renal, and respiratory function of the preterm infants. A metaanalysis found that higher doses of caffeine (10-20 mg/kg daily) were associated with greater efficacy than lower doses (5-10 mg/kg daily) and that higher doses were more closely associated with withdrawal from the ventilator, but higher doses were also more associated with tachycardia than lower doses (Chen J, Jin L, Chen X. Efficacy and Safety of Different Maintenance Doses of Caffeine Citrate for Treatment of Apnea in Premature Infants: A Systematic Review and Meta-Analysis. Biomed Res Int. 2018 Dec 24;2018:9061234. doi: 10.1155/2018/9061234.). In this study, the optimal caffeine dose cann’t be given. This can be discussed more detailed in the discussion section.

The total adverse effects were not clear and described in general. There are also studies that show long-term safety of caffeine therapy for apnea of prematurity, Mürner-Lavanchy et al reported that neonatal caffeine therapy for apnea of prematurity improved visuomotor, visuoperceptual, and visuospatial abilities at age 11 years. General intelligence, attention, and behavior were not adversely affected by caffeine (Mürner-Lavanchy IM, Doyle LW, Schmidt B, Roberts RS, Asztalos EV, Costantini L, Davis PG, Dewey D, D'Ilario J, Grunau RE, Moddemann D, Nelson H, Ohlsson A, Solimano A, Tin W, Anderson PJ; Caffeine for Apnea of Prematurity (CAP) Trial Group. Neurobehavioral Outcomes 11 Years After Neonatal Caffeine Therapy for Apnea of Prematurity. Pediatrics. 2018 May;141(5):e20174047. doi: 10.1542/peds.2017-4047.) Long-term effects of caffeine therapy can be specified in the total adverse effects discussion section.

Further controlled studies are needed to confirm the comparative usefulness of caffeine.

6. PLOS authors have the option to publish the peer review history of their article (what does this mean?). If published, this will include your full peer review and any attached files.

Reviewer #1: No

Reviewer #2: No

---

## [Author Response · Author response to Decision Letter 0]

16 Aug 2022

Dear Editors:

Thank you for providing us with such a great opportunity to submit a revised version of our manuscript. We appreciate your warm help with our manuscript. Meanwhile, we would like to express our sincere gratitude to all reviewers for their detailed and constructive comments on our manuscript. According to those helpful suggestions, we have revised the manuscript to make our results convincing. Revised portions are marked in red on the paper. The main corrections in the paper and the responses to the reviewers' comments are as follows. 

At the same time, we ensure that the manuscript meets PLOS ONE's style requirements. The corresponding author's ORCID ID is 0000-0001-9144-2805, we've validated and updated it in ‘Update my Information’. We mentioned the retrieval date in the abstract and method section: Methods: PubMed, Scopus, Embase, EBSCO, Web of Science, and Cochrane databases were systematically searched from May 1975 to June 2022 (Page2 line34);We searched Embase, PubMed, Web of Science, EBSCO, Cochrane, and Scopus databases from May 1975 to June 2022 for all relevant articles(Page4 line87). At your suggestion we have changed the title to “Comparative efficacy and safety of caffeine citrate and aminophylline in treating apnea of prematurity: A systematic review and meta-analysis”. In addition, we have checked all the references to ensure that they are complete and correct.

Thank you again for your time and help with our manuscript. We hope you will be satisfied with the revised version and look forward to hearing from you.

Sincerely,

The authors

Encl. Responses to the comments from Reviewer 1 and 2.

Reply to Reviewer #1

Dear Reviewer,

Thank you very much for your time spent in reviewing our manuscript and for your encouraging comments on its merits. After careful consideration, we have further revised the article. We hope that you will be more satisfied with the revised version.

Comments:

“Methylxanthines, including caffeine citrate and aminophylline, is the most common pharmacologic treatment for apnea of prematurity. However, due to the lack of high-quality evidence, there are no clear suggestions or guidelines on how to choose between caffeine and aminophylline. This meta-analysis aimed to assess the comparative efficacy and safety of caffeine and aminophylline for AOP, and provide reliable evidence for clinical medication in the treatment for apnea of prematurity. This meta-analysis reveals that caffeine citrate and aminophylline have similar therapeutic effectiveness on respiratory function, but caffeine has fewer side effects and should be considered first for treatment.

So this research is a nice work.”

Thank you very much for your affirmation of this article. We hope you will find this revised version more satisfactory. We are more than happy to make any further changes that will improve the article and facilitate successful publication.

Sincerely,

The Authors

Reply to Reviewer #2

Dear Reviewer,

We are really grateful to you for your time and efforts put into the comments. Those comments are all valuable and helpful for revising and improving our paper. We have studied all comments carefully and have made conscientious corrections. We appreciate your clear and detailed feedback and hope that the explanation has fully addressed all your concerns. 

Comments1:

The study is well designed, writing and grammar are clear.

There were limited RCTs in the study and it’s difficult to recommend the optimal treatment.

Early caffeine initiation is recommended in the literature.

In a study, early caffeine initiation (initial dose on the 1st day of life) was associated with a decreased need for invasive ventilatory support and lower incidence of IVH and PDA (PLoS ONE 12(12): e0189152. https://doi.org/10.1371/journal.pone.0189152X) that they recommend the initiation of caffeine treatment in the first 24 hours of life. The initiation time of caffeine treatment and its relationship with adverse effects can be mentioned in the discussion section.

Response 1:

Thank you for your good advice. Your suggestions really mean a lot to us. After careful consideration, we have added relevant content in the discussion section: Some studies have further shown that prophylactic use of caffeine for preterm infants at risk of apnea reduced the duration of mechanical ventilation[31]. Borszewska-Kornacka[32] found that preventive use of caffeine within 24 hours after birth can reduce the use of invasive ventilator and the incidence of IVH and PDA compared with therapeutic use of caffeine. However, whether caffeine should be used prophylactically in all preterm infants remains controversial. (Page8 line221)

Comments 2:

The dose of caffeine treatment can be challenging because of the limited hepatic, renal, and respiratory function of the preterm infants. A meta analysis found that higher doses of caffeine (10-20 mg/kg daily) were associated with greater efficacy than lower doses (5-10 mg/kg daily) and that higher doses were more closely associated with withdrawal from the ventilator, but higher doses were also more associated with tachycardia than lower doses (Chen J, Jin L, Chen X. Efficacy and Safety of Different Maintenance Doses of Caffeine Citrate for Treatment of Apnea in Premature Infants: A Systematic Review and Meta-Analysis. Biomed Res Int. 2018 Dec 24;2018:9061234. doi: 10.1155/2018/9061234.). In this study, the optimal caffeine dose cann’t be given. This can be discussed more detailed in the discussion section.

Response 2:

Thanks for your valuable comments. Your suggestions are really important to us. Through thinking and consulting literature, we have added relevant content in the discussion section: In addition, the dosage of caffeine therapy can be challenging because of the limited hepatic, renal, and respiratory function of preterm infants. A previous study has shown that high doses of caffeine lead to the decrease of mesenteric blood flow and serious feeding intolerance of caffeine generally appears at concentrations >60 mg/L[30]. But chen[38] found that higher doses of caffeine (10-20 mg/kg daily) were more effective than lower doses (5-10 mg/kg daily) and that higher doses were more closely associated with withdrawal from the ventilator. (Page9 line248)

Comments3:

The total adverse effects were not clear and described in general. There are also studies that show long-term safety of caffeine therapy for apnea of prematurity, Mürner-Lavanchy et al reported that neonatal caffeine therapy for apnea of prematurity improved visuomotor, visuoperceptual, and visuospatial abilities at age 11 years. General intelligence, attention, and behavior were not adversely affected by caffeine (Mürner-Lavanchy IM, Doyle LW, Schmidt B, Roberts RS, Asztalos EV, Costantini L, Davis PG, Dewey D, D'Ilario J, Grunau RE, Moddemann D, Nelson H, Ohlsson A, Solimano A, Tin W, Anderson PJ; Caffeine for Apnea of Prematurity (CAP) Trial Group. Neurobehavioral Outcomes 11 Years After Neonatal Caffeine Therapy for Apnea of Prematurity. Pediatrics. 2018 May;141(5):e20174047. doi: 10.1542/peds.2017-4047.) Long-term effects of caffeine therapy can be specified in the total adverse effects discussion section.

Further controlled studies are needed to confirm the comparative usefulness of caffeine.

Response 3:

Thanks for your valuable comments. It's a great help for us to have your suggestions. Through consideration and literature review, we modified the problem in the revised manuscript: Liu[42] has shown that caffeine may have a direct long-term neuroprotective effect on the developing brain. Mürner-Lavanchy et al[43] also reported that caffeine therapy for apnea of prematurity improved visuomotor, visuoperceptual, and visuospatial abilities at age 11 years. General intelligence, attention, and behavior were not adversely affected by caffeine. In order to truly prove the safety of caffeine citrate, more long-term follow-up studies are necessary.(Page10 line274)

Thank you very much for your affirmation and suggestions on this article. We hope the revised version will make you even more satisfied. If there is anything that needs to be modified, please do not hesitate to point it out. We are more than happy to make any further changes that improve the article.

Sincerely,

The Authors

---

## [Editor Report · Decision Letter 1]

7 Sep 2022

Comparative efficacy and safety of caffeine citrate and aminophylline in treating apnea of prematurity: A systematic review and meta-analysis

PONE-D-22-17592R1

Dear Dr. Aihua Wang

We’re pleased to inform you that your manuscript has been judged scientifically suitable for publication and will be formally accepted for publication once it meets all outstanding technical requirements.

Kind regards,

Hasan Tolga Celik, M.D.

Academic Editor

PLOS ONE

---

## [Editor Report · Acceptance letter]

9 Sep 2022

PONE-D-22-17592R1 

Comparative efficacy and safety of caffeine citrate and aminophylline in treating apnea of prematurity: A systematic review and meta-analysis 

Dear Dr. Wang:

I'm pleased to inform you that your manuscript has been deemed suitable for publication in PLOS ONE. Congratulations! Your manuscript is now with our production department. 

Kind regards, 

on behalf of

Dr. Hasan Tolga Celik 

Academic Editor

PLOS ONE